

# Combining statistical and hydrodynamic models to assess compound flood hazards from rainfall and storm surge: a case study of Shanghai

Hanqing Xu[1,2], Elisa Ragno[2], Sebastiaan N. Jonkman[2], Jun Wang[1], Jeremy D. Bricker[2,5], Zhan Tian[3], Laixiang Sun[4]

[1] Institute of Eco-Chongming (IEC), Key Laboratory of Geographic Information Science of Ministry of Education, Shanghai Key Lab for Urban Ecological Processes and Eco-Restoration, Institute for National Safety and Emergency Management, East China Normal University, Shanghai 200241, China
[2] Faculty of Civil Engineering and Geosciences, Delft University of Technology, Delft 2628CN, The Netherlands
[3] School of Environmental Science and Engineering, Southern University of Science and Technology, Shenzhen 518055, China
[4] Department of Geographical Sciences, University of Maryland, College Park, USA
[5] Department of Civil and Environmental Engineering, University of Michigan, Ann Arbor, MI 48109, United States

*Correspondence to*: Hanqing Xu (h.xu-4@tudelft.nl)

**Abstract.** Coastal regions have experienced significant environmental changes and increased vulnerability to floods caused by the combined effect of multiple flood drivers such as storm surge, heavy rainfall, and river discharge, i.e., compound floods. Hence, for a sustainable development of coastal cities, it is necessary to understand the spatiotemporal dynamics and future trends of compound flood hazard. While the statistical dependence between flood drivers, i.e., rainfall and storm surges, has been extensively studied, the sensitivity of the inundated areas to the relative timing of driver's individual peaks is less understood and location dependent. To fill this gap, here we propose a framework combining a statistical dependence model for compound event definition and a hydrodynamic model to assess inundation maps of compound flooding from storm surge and rainfall during typhoon season in Shanghai. First, we determine the severity of the joint design event, i.e., peak surge and precipitation, based on the copula model. Second, we use the Same Frequency Amplification (SFA) method to transform the design event values in hourly timeseries so that they represent boundary conditions to force hydrodynamic models. Third, we assess the sensitivity of inundation maps to the time lag between storm surge peak and rainfall. Finally, we define flood zones based on the primary flood driver and we delineate flood zones under the worst compound flood scenario. The study highlights that the temporal delay between storm surge and rainfall plays a pivotal role in shaping the dynamics of flooding events. More specifically, the worst conditions in terms of cumulative inundation depth occur when rainfall precedes the storm surge peak. At the same time, the results show that in Shanghai surge is the primary flood driver. High storm surge at the eastern part of the city (Wusongkou tidal gauge) propagate upstream in the Huangpu River resulting in fluvial flooding in Shanghai city center and several surrounding districts. This calls for a better fluvial flooding control system hinging on the backwater effect during high surge in the upper and middle Huangpu River and in the newly added urbanized areas to ensure flood resilience. The proposed framework is useful to evaluate and predict flood hazard in coastal cities, and the results can provide guidance for urban disaster prevention and mitigation.



# 1 Introduction

Despite significant advancements in flood management systems (e.g., forecasting and early warning systems), recent compound flood events such as Hurricane Harvey (2017) and Super Typhoon Lekima (2019), have demonstrated the vulnerability of coastal cities (Zscheischler et al., 2018; Hendry et al., 2019; Valle-Levinson et al., 2020). Compound flood events are defined as events generated by the combined effect of multiple physical drivers, e.g., heavy rainfall and storm surge. Due to the interaction between flood drivers, the magnitude of compound events can be greater than the magnitude of events generated by a single driver, e.g., either heavy rainfall or storm surge. Hence, it is important to understand how such interaction is realized in coastal areas prone to compound floods.

According to the Intergovernmental Panel on Climate Change (IPCC), adapting coastal cities to the adverse impacts of climate change is crucial for their long-term resilience and sustainability (Adler et al., 2022). Consequently, a comprehensive assessment of the processes leading to compound floods is crucial to develop targeted and innovative adaptation strategies.

Chinese southeastern coastal cities are highly susceptible to flood hazards caused by typhoons due to their unique geographical location and complex ecological environment (Xu et al., 2023). During the occurrence of typhoon, the simultaneous presence of heavy rainfall and strong storm surge can result in severe and destructive flood events (Wahl et al., 2015; Santiago-Collazo et al., 2019; Ridder et al., 2020). In this context, the effective management of flood risk relies on the integration of statistical and hydrodynamic modeling to assess the extent and potential impact of a flooding event.

In recent years, a growing body of research has focused on the probabilistic characterization of compound flooding (Moftakhari et al., 2019; Gori et al., 2020; Xu et al., 2023). Bevacqua et al. (2019) assessed the potential probability of compound flood hazards triggered by heavy precipitation coinciding with high tidal level across Europe. Gori et al. (2020) simulated a large number of realistic TC events to examine TC flooding driven by rainfall and storm surge.

At the same time, hydrodynamic models have been widely used to simulate flood extent due to single or multiple flood drivers (Yin et al., 2016; Wang et al., 2018; Shi et al., 2020). Hydrodynamic models are effective in simulating the consequences of rainfall-runoff and storm surge during typhoon events (Kumbier et al., 2018; Bevacqua et al., 2019; Zellou and Rahali, 2019) and can improve our understanding of region-specific hydrodynamics and the genesis mechanisms of compound flooding scenarios (Xu et al., 2022). Such numerical models necessitate boundary conditions, which include hourly timeseries of sea water level and rainfall data, to ensure precise and effective simulations of inundation extent and depth. Paprotny et al. (2017) combines Bayesian-network-based model and one-dimensional steady-state hydraulic model in place of conventional rainfall-run-off models to carry out flood mapping for Europe. Moftakhari et al. (2019) proposed a method linking bivariate statistical models and hydrodynamic models to estimate compound floods resulting from upstream discharge and downstream water levels in tidal estuaries. Gori and Lin (2022) conducted an investigation into compound flood hazards under climate change by combining physical models and joint probability analysis, highlighting its effectiveness in reducing computational expenses.





Generally, copula models can provide information on the joint severity of multiple flood drivers, like rainfall, river discharge and storm surge, which needs to be further processed to become hydraulic boundary conditions. However, in case of multiple flood drivers (e.g., surge and rainfall), it is essential to consider not only the statistical dependence of the drivers,

but also the physical interactions between them in terms of flooding processes. This comprehensive approach is crucial for a more accurate assessment of compound flood and the development of effective flood management strategies. For example, when high sea levels from storm surge coincide with  increased river water flow due to heavy rainfall, it would lead to backwater effects for drainage and result in localized flooding. This implies the need to investigate methods to move from the definition of the severity of the event to the physical interaction of the flood drivers at a sub-daily time scale.

Moreover, when sub-daily observations are available to force a hydrodynamic model, to obtain a better understanding of generation process of compound flood, it is necessary to investigate the sensitivity of the flood extent to the relative timing between the drivers, i.e., between storm surge peak and rainfall. This enables a clear identification of inundation zones and their respective drivers, which are key information especially in urban planning to improve the resilience of the system. Currently, there is limited research on how to clearly define flood zones in coastal cities based on their main flood sources

(Jane et al., 2020; Muñoz et al., 2021) since a systematic numerical analysis is required to delineate flood zones, which are very sensitive to the relative occurrence of the tidal fluctuation and the rainfall event.

The objective of this paper is to combine dependence models for probabilistic characterization of compound events with hydrodynamic models to improve the identification of flood zones prone to compound floods. To assess flood hazards and the associated main physical drivers we propose the following framework: (1) determine events of surge and rainfall from

their the joint probability of occurrence; (2) select historical events to derive sub-daily signal of surge and rainfall; (3) merge (1) and (2) via the SFA method to generate boundary conditions; (4) simulate flood zones for various time lags between peak surge and rainfall to enable adaptation strategies specific to the type of flood driver(s). A comprehensive urban hydrodynamic inundation model is used to assess the sensitivity of the inundation maps to the relative timing between peak surge and rainfall. This quantitative assessment methodology of compound flooding can assist engineers and urban planners

in designing resilient structures in flood-prone areas.





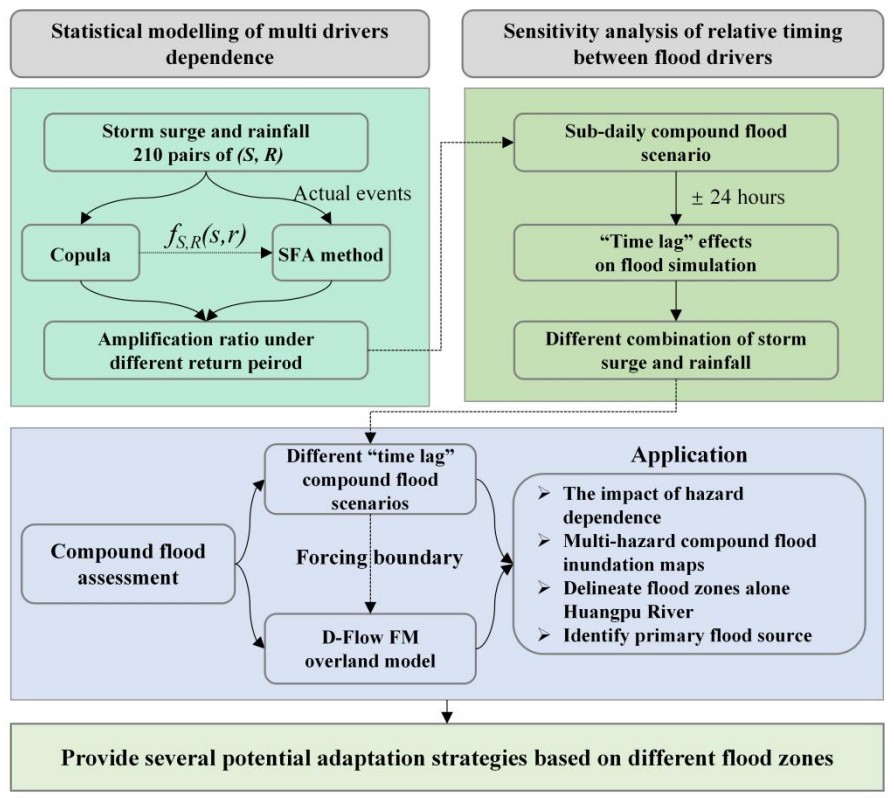

**Figure 1: Flowchart of this study.**

## 2 Materials

### 2.1 Study area

Shanghai, a coastal megacity located in Yangtze River delta, spans a total area of 6,340.5 km2 and had a population of 24.87 million in 2020 (Figure 2). The city experiences an annual rainfall of approximately 1,200 mm, with the rainy months falling between June and September. During this period, Shanghai is prone to frequent typhoons and rainstorms, which often result in storm flooding, leading to severe damage. In 1997, Typhoon Winnie caused economic damage exceeding US $100 million. Despite the presence of Shanghai's extensive flood mitigation measures (e.g., 523-kilometer seawall, comprehensive

drainage systems and advanced water level monitoring systems), the city experienced significant impacts during Typhoon Winnie in 1997 (killed 342 people and caused a direct economic loss of USD 4.3 billion). These instances highlight Shanghai's persistent vulnerability to extreme weather events.



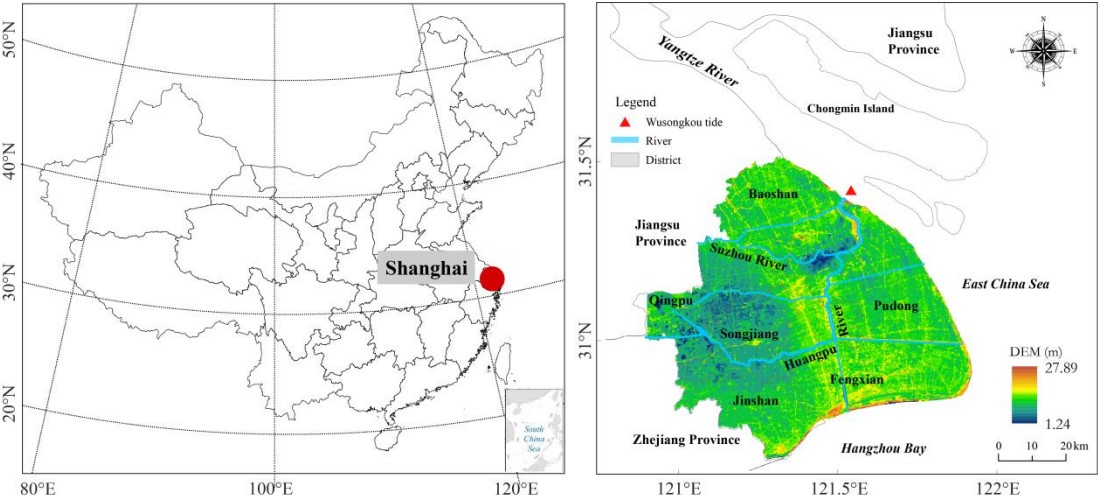

**Figure 2: Location map of the study area and digital elevation map of Shanghai (red dot in left graph is the location of Shanghai in China. The red triangle in the right graph shows Wusongkou tide gauge. The DEM data use the Wusong datum).**

## 2.2 Dataset

Two flood drivers are considered in this study: storm surge peaks and associated daily rainfall. Daily rainfall is collected from the China Meteorological Administration (CMA, http: //data.cma.cn/) Meteorological Data Center, covering the period from 1961 to 2018. More specifically, the reanalysis dataset of storm surge peaks from 1961 to 2018 located at the mouth of the Huangpu River (Figure 2) and associated rainfall used here is the same as the one described and used in Xu et al. (2022) and the reader is referred to that paper for further details on data processing. To obtain hourly boundary conditions, a representative hourly storm surge pattern and hourly rainfall hyetograph are selected from the Shanghai Municipal Water Authority and CMA.

In addition to flood drivers, we use land use data to analyze the potential impact of compound floods. Our approach involves directly coupling the inundation maps with land use data in order to identify the potential impacts, without employing a separate damage function. In this study impacts are thus quantified in terms of the affected area for different land use types in different flood zones. The land use type data of Shanghai in 2010 comes from the Shanghai Institute of Surveying and Mapping. This represents the most detailed data available for Shanghai; however, it's important to recognize that land use patterns and urban development may have evolved since then. The land use type are further divided into 4 categories, which are residential land (including urban residential land and rural homestead), transportation and public service land (including commercial service land, public management and railway, highway, port and other transportation facilities), industrial land, green and agricultural land (including farmland, garden land, woodland and park green space).





## 3 Methods

### 3.1 Same Frequency Amplification for generating boundary conditions

Drawing on the concept of an amplifier, we implement the Same Frequency Analysis (SFA) method to establish boundary conditions for hydrodynamic modelling (Xiao et al., 2009). The SFA method amplifies observed events ($v_0(t)$) to meet the design criteria ($v_i(t)$) via a constant amplification factor (A), then $v_i(t) = A \cdot v_0(t)$. In this specific case study, hourly storm surge and hourly hyetograph over a window of 24 hours are amplified based on predetermined design events expressed in terms of storm surge peaks and total daily rainfall.

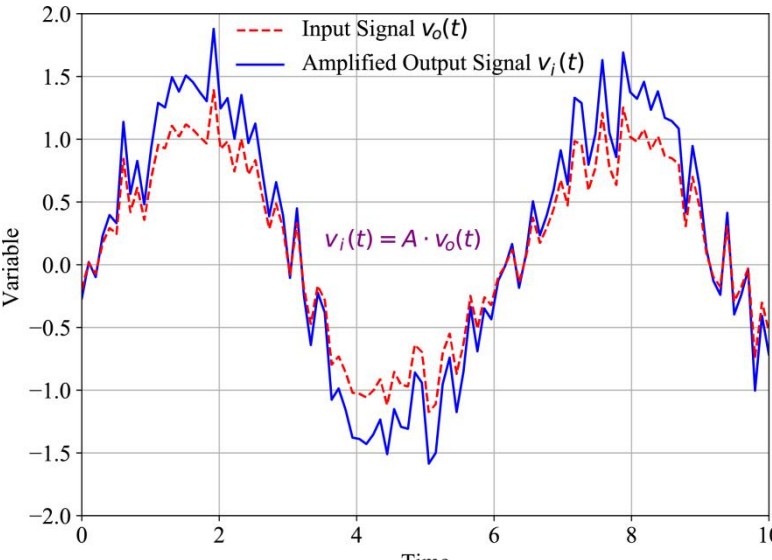

**Figure 3: Principal of the Same Frequency Analysis (SFA) based on the principal of signal amplification.**

These design values derived from the probabilistic dependence model presented in Xu et al., 2022. Readers are referred to (Xu et al., 2022) for details on the selection of the severity and magnitude of the event of interest.

Since design values are single values, to obtain hourly storm surge and rainfall time series the following steps are considered.

First, a local typical storm surge pattern and rainfall hyetograph over a window of 24 hours are selected from historical observations. According to the peak surge and cumulative rainfall volume obtained from the joint probability model, we then calculate the amplification ratio by dividing the peak of the typical surge pattern and cumulative value of rainfall by the corresponding design event, Figure 4. The amplification factor is estimated by dividing the design value of the peak surge or cumulative rainfall signal from the typical storm event. By applying this amplification factor, we ensure the amplification

ratio essentially adjusts the typical storm event to match the desired design criteria based on the design value from the copula function.

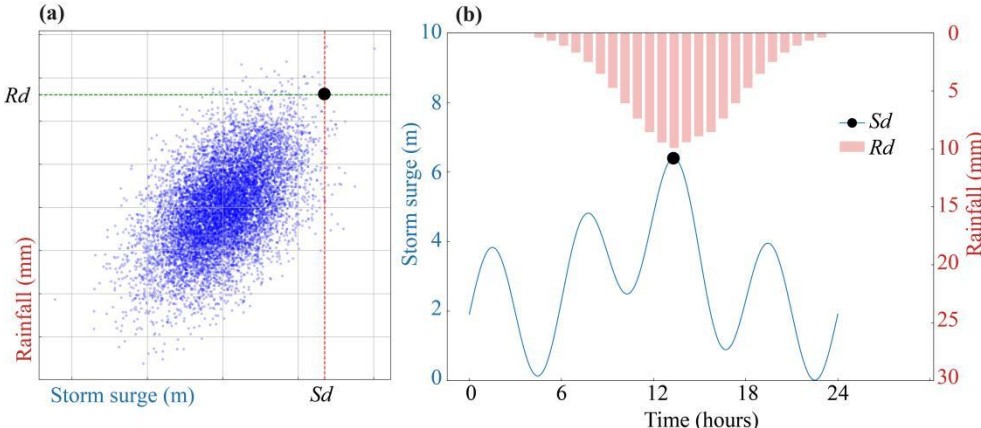

**Figure 4: Sketch graph to illustrate the combination of the copula and SFA method. Panel (a) represents the dependence model of surge peaks and accumulated rainfall to derive their respective design values $S_d$ and $R_d$. Panel (b) illustrates the amplified signals where the highest peak corresponds to the design value $S_d$ and the volume of rainfall corresponds to the design value $R_d$.**

### 3.2 Hydrodynamic overland inundation model

D-Flow FM, developed by Deltares in the Netherlands, has been widely applied for water flow simulations, including storm surge and overland inundation, because of its capability of simulating 2D and 3D shallow water flow. It integrates structured and unstructured grids, which is convenient for partial grid refinement and predict the behavior of water flow for flood modeling in coastal regions (Deltares, 2018). Ke et al. (2021) developed a 2D urban model for Shanghai with D-Flow FM capable of reproducing coastal, fluvial, pluvial and interacting flood dynamics. Hence, we use the D-Flow FM hydrodynamic model, equipped with flexible meshing, efficient numerical schemes and comprehensive physical equations, to simulate overland flood inundation maps. The model used in this study incorporates rainfall and storm surge boundary conditions and takes into accounts floodwalls along the Huangpu River in Shanghai, as developed and validated by Ke et al. (2021). We use the digital elevation model data acquired from the Institute of Geographic Sciences and Natural Resources Research, Chinese Academy of Sciences. River, flood wall and discharge data obtained from the Shanghai Municipal Water Authority, to develop the urban inundation model. The overland flooding model combines regular rectangular and irregular triangular meshes. This model is a surface runoff numerical model, and the mesh grid resolution is set as 150 m. The storm surge at Wusongkou gauge is used as the downstream boundary condition for the overland inundation model. The hourly rainfall hyetograph data from the CMA as input to the overland inundation model to determine the spatial and temporal distribution of rainfall across Shanghai.

### 3.3 Definition of different flood zones

To discern the principal flood contributor, namely storm surge and/or precipitation, for a particular region, Bilskie and Hagen (2018) propose a floodplain division into three zones based on the dominant contributor: fluvial (surge), transition (surge and precipitation) and pluvial (precipitation) zones. Here, we are interested in defining such zones in the city of





Shanghai since they represent key information to develop targeted adaptation strategies. Hence, we run the hydrodynamic model for three different cases (Table 1). In Case 1, only the tidal level time series is taken as input, with no rainfall, to identify fluvial zones. In Case 2, rainfall and a fixed tidal level of 0 meters are taken as the boundary condition to identify pluvial zones. In Case 3, both storm surge and rainfall runoff processes are considered to identify transition zones. We expect that areas along the river flood plain are more prone to flooding due to high surge and backwater effect. On the other hand, we expect that inland areas are prone to pluvial flooding since rainfall becomes the predominant contributor to flooding.

Table 1. Cases used in flood zone delineation.

| Case | Inundation types | Description of modelling approach |
|---|---|---|
| Case 1 | Fluvial zones | Tidal level without rainfall |
| Case 2 | Pluvial zones | rainfall with a fixed tidal level of 0 meter |
| Case 3 | Transition zones | Tidal level and corresponding return period rainfall |

## 4 Results

### 4.1 Hourly boundary conditions merging copulas and SFA

In this study, we use the hourly storm surges curve at Wusongkou tidal gauge and rainfall hyetograph in Shanghai recorded during Typhoon Winnie (9711), between 18th August to 19th August, as the basic curves to construct boundary conditions. This specific event stands out as it represents a noteworthy combination of extreme storm surge and heavy rainfall. Typhoon Winnie brought historical records of storm tide of 5.99 m at Wusongkou tide gauge and 134.3 mm rainfall in a period of 24 hours. Since this event is representative of a combination of extreme storm surge and heavy rainfall, it is suitable for studying the impacts of compound floods.

The amplification factor for storm surge and rainfall is estimated so that the amplified storm and rainfall match the design values derived from the probabilistic dependent model presented in Xu et al. (2022) and shown in Figure 5. Since Shanghai is generally more vulnerable to fluvial flooding, we select the joint design values such that the surge design value corresponds to the event with a return period of 1000 years (the design standard of flood wall alone middle and lower reaches of Huangpu River) when yearly maximum water level from 1912 to 2013 are analyzed independently (Ke et al., 2018). The associated rainfall design value is obtained from the dependence model, Figure 5(a). By fixing the design value of the surge peak, the pairs of surge and rainfall with the highest density corresponds to a joint event with a return period of 200 years considering the "AND" scenario, i.e., the probability that both surge and rainfall are higher than their respective



thresholds is 0.005 (Figure 5(a)). The SFA is then applied to the surge and rainfall during Typhoon Winnie and amplified

based on the designed value identified (Figure 5b).

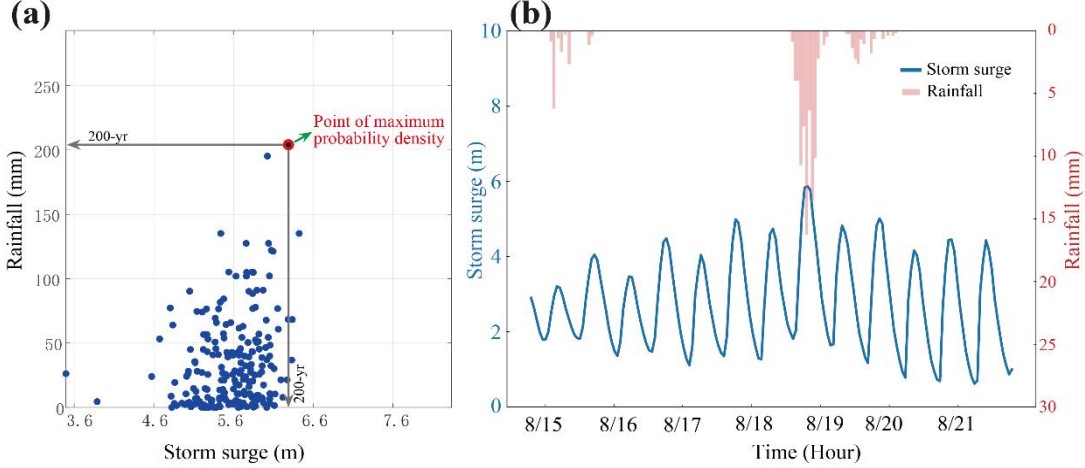

**Figure 5: (a) Bivariate statistical analysis of joint distribution with "AND" scenario between peak storm surge and corresponding cumulative rainfall by MvCAT; (b) The boundary conditions for the hydrodynamic model based on a scenario of no time lag**
**between the peak storm tide and the corresponding cumulative rainfall observed during typhoon Winnie.**

## 4.2 Sensitivity of inundation maps to time lag between hourly peak storm surge and rainfall

To understand the combined effect of storm surge and rainfall, it is important to investigate the relative timing between the

peaks of these two drivers. Hence, we investigate how the relative timing of the peak surge and rainfall affect flood

inundation depth by running the D-Flow FM hydrodynamic model multiple times, each time with a pre-defined distance

between the peaks spanning over a temporal window of ±24 hours. For each run, the averaged inundation depth over all

flooded locations (here labelled as the cumulative inundation depth) is recorded.





**Figure 6: Cumulative inundation depth under combined impact of storm surge and rainfall with -2 and +12 hour lags. (a) The boundary conditions of -2h time lag between the peak storm surge and the corresponding cumulative rainfall; (b) Flooding**





**inundation under the -2h time lag scenario. (c) The boundary conditions of the scenario with +12h time lag between the peak storm surge and the corresponding cumulative rainfall; (d) Flooding inundation under the +12h time lag scenario. (e) Cumulative inundation depth with different time lags. (the horizontal dashed line is the maximum cumulative inundation depth for uniform rainfall. The red dot means the maximum cumulative inundation depth by -2 hours lag of compound event).**

We observe that maximum cumulative inundation depth occurs when the rainfall event occurs 2 hours prior to the surge peak

and considered. In this scenario, floods mainly occur in the Upper Huangpu River, the maximum cumulative inundation depth exceeds 1.0 m in the southwestern Qingpu district and the southern Songjiang district, while some areas in the downtown area are covered by flood water with depths over 2 m (Figure 6(a-b)). The water level in rivers are high, potentially exceeding the capacity of drainage systems and pumping infrastructure to mitigate the inundation. This results in more extensive flooding in low-lying areas. In contrast, when rainfall takes place after the surge peak (e.g. +12 hours later),

the water levels in the river have time to recede. This allows for improved drainage and pumping capacity in the city, reducing the cumulative inundation depth to 0.21 m (Figure 6(c-d)).

Generally, we observed a cumulative inundation depth greater than 0.25 m when the rainfall event precedes the peak of the surge, i.e., between -24 to 0 hours. Rainfall events occurring after the storm surge peak ( > 0 hours) do not increase the cumulative inundation depth (Figure 6e). The cumulative inundation depth stabilizes at around 0.21 m when rainfall peaks

about 12 hours after the surge peak. Therefore, the observed difference in flooding between these scenarios can be attributed to the combination of high water level in rivers following the surge peak and the capacity of the drainage and pumping systems. This highlights the critical importance of considering the relative timing of events when determining the flood extent and severity in fluvial urban region.

## 4.3 Flood zones

The distribution of flood zones would vary significantly given different storm events due to the different timing between rainfall and storm surge; here, we investigate the average response of the basin for the worst cumulative inundation depth event (rainfall peaks 2 hour prior the surge peak). Flood zones are delineated in the entire study area based on the criteria established (Table 1).

In general, storm surges are the primary cause of inundation within and along the Huangpu River (represented by red), while

the areas situated far from the river, mostly inland, experience inundation primarily due to rainfall (represented by blue). Distinct transitional zones, marked in yellow in Figure 7, can be seen surrounding the Huangpu River. We observe three main critical areas (Region A, B and C) due to the obvious cumulative inundation depth. Region A (inundated by 39.68 km$^2$), with its immediate fluvial adjacency, dominated by fluvial flooding. Similarly, Region C (inundated by 23.88 km$^2$) is also dominated by fluvial flooding even though located about 100 km from the coast. This could be due to the river configuration

and the backwater effect that prevents the normal discharge of the river into the sea when high surge occurs. In contrast, Region B (inundated by 34.27 km$^2$) is the most prone to the combined effect of rainfall and storm surge with the largest transition zone recorded, 2.3 times greater than the transition zone in Region A and 4.4 times greater than the transition zone



in Region C. The combination of low elevation and low flood wall makes the Region B susceptible to fluvial flooding, which can result in overtopping and further exacerbate flooding.

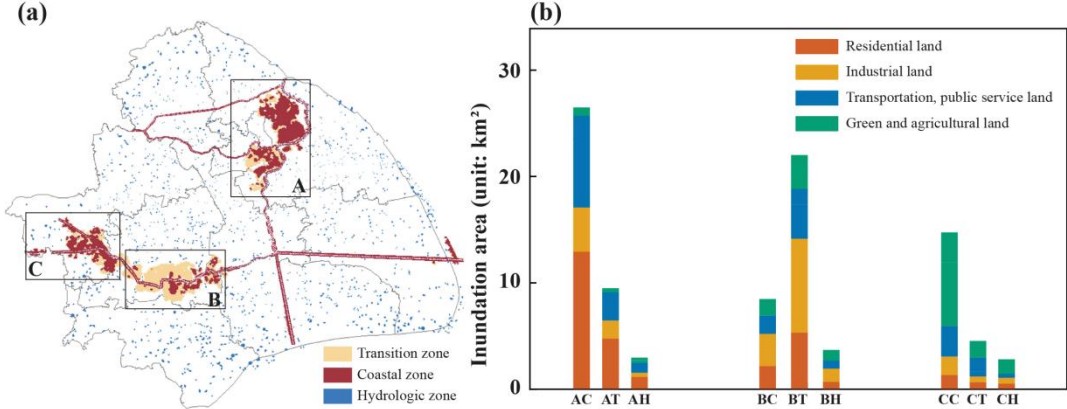

**Figure 7: (a) Flood zones under 200-yr return period flooding events; (b) The inundation area of different land use combination under 200-yr return period flooding events (C: fluvial zone; T: transition zone; H: pluvial zone). The letters in front of the fluvial zones (A, B, C) refer to the region investigated.**

To gain additional insight on the population exposed in the regions identified as the most affected by a flood event, we investigate urban land use maps. Figure 7(b) clearly shows that Region A is mostly residential land, Region B is mostly industrial land and Region C is mostly green and agricultural land. This implies that a different approach is needed when developing strategies for flood control, especially in region B where the industrial sector is prone to compound flooding (a larger transition zone). These results give insights on how to effectively realize the developmental goals for the year 2035 in Shanghai's new city expansion endeavors. These results suggest that the focal point should be on reducing the risk in the system for fluvial flooding. This would protect also upper and middle reaches of the Huangpu River, as well as newly established urbanized region, with particular attention to low-lying areas in the upper reach of the Huangpu River.

## 5 Discussion

The primary objective of this study was to develop a methodology to quantify flood inundation depth accounting from the relative timing of surge and rainfall and their respective magnitudes, when the dependence between the two events cannot be neglected, such as during typhoon season. In our simulations, the time lag between rainfall and surge peak equal to -2 hours, i.e., the peak rainfall occurring 2 hours before the surge peak, causes the largest total inundation depth. However, this result depends on our initial choice of the hourly storm tide elevation and rainfall hyetograph amplified based on design requirements. The availability of higher quality data at finer temporal resolution could provide a different and more accurate insight into the sensitivity of the area to the combined effect between rainfall and storm surge. At the same time, we showed that this framework is a valuable option to generate boundary conditions when there is a lack of data.





From a design perspective, this study emphasizes the need for a comprehensive and case-specific understanding of the interaction between surge and rainfall and their relative timing to identify the most suitable adaptation strategy. For example, our results showed that storm surge protection systems can prevent flooding in upstream reaches of the Huangpu River. Indeed, the river's susceptibility to backwater effects can result in river flooding even during moderate rainfall events.

Alternatively, flood protection measures along riverbanks and within the city can also be considered but they should be developed considering storm surge and backwater effects as major drivers for high water in the river. Along with interventions to reduce the inundation due to storm surges, implementing rainwater storage and hydrograph peak reduction facilities, such as detention basins, can enhance the capabilities of the urban flood protection system and improve the city's safety, particularly in cases when high rainfall precedes high surge.

However, it is crucial not to overlook hydrological factors, including topography, land use, soil type, and vegetation cover, as neglecting these elements may lead to an inaccurate representation of flood risk. For example, comprehensive land use maps offer an effective tool for delineating flood prevention and mitigation strategies because they rule out solutions not suitable for a specific category. In this study, we combined the land use types with inundation maps, thereby establishing a more holistic understanding of the areas at risk. Within the context of this investigation, we put forth a range of prospective

adaptation strategies that can be employed by various vulnerable areas to reduce the impact of compound flooding. We underscore the paramount importance of adopting a comprehensive flood risk management approach that encompasses the entire system, while also emphasizing the necessity for integrated and coordinated measures to effectively tackle the intricate and dynamic nature of compound flooding events. Among these strategies is the proposal for constructing floodgates at the mouth of the Huangpu River, a measure designed to substantially diminish the impact of storm floods on the inner river

basin and to prevent or reduce the necessity to raise floodwalls. Insights form the completed and ongoing designs for international storm surge barriers could serve as valuable references for this investigation (Mooyaart and Jonkman, 2017). Concurrently, it is imperative to reinforce the flood wall of vulnerable sections on the upstream reach of the Huangpu River and newly urbanized regions to ensure the preservation of their designated flood control capacity until the floodgates at the river mouth are completed. Additionally, in accordance with the latest analysis of Huangpu River tide levels, it is advisable

to augment floodwalls to maintain their intended flood control capacity (Ke et al., 2018). By prioritizing these targeted efforts, Shanghai can make significant strides towards achieving its developmental aspirations while effectively managing the risk from compound flooding events.

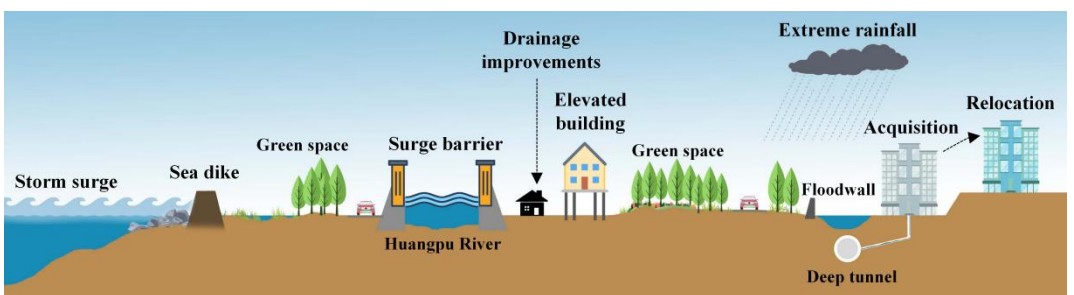




**Figure 8: Visual of various measures applicable to Shanghai.**

## 6 Conclusions

In this study, we highlight the importance of understanding how the interactions between multiple flood mechanisms change flood hazard maps, in contrast with current Chinese flood policy which focuses on floods due to only a single driver. Although this study is specific to Shanghai City, the methodology of quantitatively assessing compound flood hazard via a coupled statistical-hydrodynamic model framework is available for other coastal cities.

1) The study allowed us to analyse the dynamics of storm surge and rainfall interactions and their effect on the inundation area under various return periods.

2) The investigation into the impact of temporal windows spanning $\pm 24$ hours revealed that the maximum cumulative inundation depth occurs at a temporal lag of -2 hours, highlighting the importance of assessing compound flood hazard "time lag" between different flood driving factors.

3) Storm surge is the primary driver of fluvial flood inundation in Shanghai, with inundated regions primarily situated on both sides of the Huangpu River, and concentrated in the city center and other specific districts.

4) Based on the worst case inundation map caused by storm surge and rainfall, the flood zones are delineated into fluvial, pluvial and transition zones, enabling us to propose potential adaptation strategies to cope with inundation in Shanghai.

5) This knowledge can assist engineers and urban planners in planning and designing structural and non-structural measures in flood-prone areas to withstand potential combinations of hazards.

**Data availability.** The data used in this study such as the daily rainfall records are collected from the China Meteorological Administration (http://data.cma.cn/).

**Author contributions.** HQ, ER, ZT, and JW conceived the study. HQ, ER, and LS drafted the manuscript. HQ and ER contributed to the Copula method and boundary conditions. JB and ZT were responsible for hydrodynamic modeling. HQ and ER contributed to the sensitivity analysis of inundation. All authors commented on the manuscript.

**Competing interests.** The contact author has declared that neither they nor their co-authors have any competing interests.

**Disclaimer.** Publisher's note: Copernicus Publications remains neutral with regard to jurisdictional claims in published maps and institutional affiliations.





**Special issue statement**

This article is part of the special issue "Attributing and quantifying the risk of hydrometeorological extreme events in urban environments". It is not associated with a conference.

**Acknowledgments.** Hanqing Xu is thankful for financial support from the program of China Scholarships Council (no. 202006140040).

**Financial support.** This research has been supported by the National Natural Science Foundation in China (Grant No. 41971199), the Major Program of the National Social Science Foundation of China (Grant No. 18ZDA105), the High-level
Special Funding of the Southern University of Science and Technology (Grant No. G02296302, G02296402), the Shanghai Key Lab for Urban Ecological Processes and Eco-Restoration (SHUES2023B02) and the Fundamental Research Funds for the Central Universities.

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
