# Peer review of "Combining statistical and hydrodynamic models to assess compound flood hazards from rainfall and storm surge: a case study of Shanghai"

_Hydrology and Earth System Sciences, 2023_

## Author Comment (AC1)

**Response to Reviewer #1's Comments and Suggestions**

*The presented study looks to assess compound flood hazards in Shanghai from rainfall and storm surge events using a statistical dependence model for compound event definition and a hydrodynamic model to represent inundation. The authors in particular look at the sensitivity of inundation to the timing of storm surge peaks and rainfall, which is valuable information for flood risk management. The study is well organized and the objectives are clearly defined and well addressed. I do have a number of relatively minor issues that should be addressed before the paper is published. Please find them below:*

*1. Add quantitative results to Abstract.*
**Response:** We appreciate your suggestion. We have revised the Abstract to include quantitative findings. For example, "More specifically, that the peak rainfall occurs two hours before the peak storm surge would cause the deepest average cumulative inundation depth."

*2. "TC" events, tropical cyclone acronym should be spelt out the first time*
**Response:** We spelled out the "tropical cyclone (TC)" when it appears in the second paragraph of introduction for first time.

*3. Paragraph starting line 67 is missing some references. The example provided line 71, are the authors saying this is something that can happen in theory, or that has been documented to happen in the case study area (reference?).*
**Response:** We added relevant references (Hao et al., 2016; Jalili et al., 2022) for paragraph starting at line 67 and appropriate references (Feng et al., 2022) to support the example provided in line 71.

References:
Hao Z., and Singh VP.: Review of dependence modeling in hydrology and water resources. Progress in Physical Geography, 40(4):549-78, doi:10.1177/030913331663246, 2016.Jalili Pirani, F., and Najafi, M. R.: Multivariate analysis of compound flood hazard across Canada's Atlantic, Pacific and Great Lakes coastal areas. Earth's Future, 10(8), e2022EF002655, doi:10.1029/2022EF002655, 2022.Feng, D., Tan, Z., Engwirda, D., Liao, C., Xu, D., Bisht, G., Zhou, T., Li, H.-Y., and Leung, L. R.: Investigating coastal backwater effects and flooding in the coastal zone using a global river transport model on an unstructured mesh, Hydrol. Earth Syst. Sci., 26, 5473–5491, https://doi.org/10.5194/hess-26-5473-2022, 2022.

*4. Line 135. A little more detail on what is meant by a "local typical storm surge pattern and rainfall hyetograph" would be beneficial. What is a typical storm and how is it defined?*
**Response:** We provided additional details to explain the concept of a "local typical storm surge pattern and rainfall hyetograph," which refers to historical patterns

observed in the Shanghai region. This pattern is characterized based on past storm events, notably during the "9711" typhoon event in August 1997. During this event, the sea level rose significantly, causing high tide levels in the Huangpu River to break historical records. At the Mishidu and Damaogang stations, both located upriver from Shanghai, water levels were 24 cm and 26 cm higher than their previous historical records, respectively. The peak tide occurred just after the heaviest rainfall.

*5. a follow up remark is that total sea level during a surge is referred to as "storm surge" in the paper, including in the result figures. How are tides considered in this study? Is the surge residual set at high tide?*

**Response:** Thank you for pointing this out. We have clarified that the total sea level during a surge, referred to as "storm surge" in the paper and shown in the result figures, includes the tidal component. The storm surge residual is set at high tide to represent the worst-case scenario. If the reference is mean high water instead of mean sea level, the length of the tide part bars may be smaller. However, we think that the peak water level, driven by the combination of astronomical tide, storm surge, and relative sea level rise, poses the most significant hazard to coastal cities.

*6. Bilskie and Hagen (2018) defines zones as "coastal", "hydrologic" and "transition". The use of the term "fluvial" here to refer to surge-induced only might be misleading as fluvial flooding in its typical definition can be rainfall-induced. Unless a clearer explanation is provided, "Tidal zone" or "coastal zone" might be more accurate. Jumping forward, the Bilskie and Hagen terminology is used in Figure 7 so there needs to be consistency on this throughout the paper.*

**Response:** Thank you for your suggestion. In this study, we focus on the water level changes at the outlet of the Huangpu River (Wusongkou tide gauge), without considering the coastal region over the sea. Initially, we used the term "fluvial" to refer to surge-induced flooding. To avoid any confusion, we clarify in the article that "fluvial" in this context does not refer to rainfall-induced flooding, but rather to the water level changes caused by storm surges. We have updated Figure 7 to use "fluvial zones" and "pluvial zones" instead of "coastal zones" and "hydrologic zone".

*7. How was the hydrodynamic model calibrated and validated? I could not find Ke et al. (2021) in the references.*

**Response:** We apologize for the oversight. The overland flood model covers the mainland of Shanghai city with triangular cells of typical dimension 150 m and curvilinear grid cells cover the Huangpu River and other major canals. In the revision, we provided additional information about the hydrodynamic model and added Ke et al. (2021) in the references.

Refereces:
Ke, Q., Yin, J., Bricker, J., Savage, N., Buonomo, E., Ye, Q., Visser, P., Dong, G., Wang, S., Tian, Z. and Sun, L.: An integrated framework of coastal flood modelling under the failures of sea dikes: a case study in Shanghai. Nat. Hazards, 109(1),

671-703, doi:10.1007/s11069-021-04853-z, 2021.

*8. Figure 6 b and d, add scale and north arrow.*
**Response:** Thank you for this suggestion, we added the scale and a north arrow to Figure 6(b) and (d).

*9. Line 229: "represented by red" in Figure 7?*
**Response:** Thank you for pointing out this unclear sentence. We changed this part as follows: "In general, storm surges are the primary cause of inundation along and within the Huangpu River, as indicated by the areas marked in red in Figure 7. The inland areas situated far from the river experience inundation primarily due to rainfall, shown in blue. Distinct transitional zones, marked in yellow in Figure 7, can be seen surrounding the Huangpu River."

*10. While the discussions provide insights on the study's limitations and its wider implications on flood mitigation, there is a lack of references to past studies that have looked at flooding risk in Shanghai and how the author's findings relate to them.*
**Response:** We added four additional references to three previous studies related to flooding risk in Shanghai in the discussion section and improved the links between our findings and the existing literature.

---

## Author Comment (AC2)

**Response to Reviewer #2's Comments and Suggestions**

1. *please add protection information along the Huangpu River and coastlines.*
**Response:** We added information about flood protection infrastructure along the Huangpu River in the revised figure 2 and labeled the height of flood walls.

2. *There is little inundation along the coastline, is it because of coastal protection?*
**Response:** In this study, we focus specifically on compound flooding along the Huangpu River and investigate water level changes at the outlet of the Huangpu River (Wusongkou tide gauge). We do not consider the coastal region over the sea. However, we acknowledge the presence of protective structures along the coastline, which mitigate the impact of storm surges and reduce the extent of coastal flooding. This aspect is clarified in the manuscript, particularly in the discussion section.

3 *Line 155-160, the description of how to develop a flood inundation model looks quite simple, but it is a complicated process and did not describe how to validate. I suggest adding more information here, at least, to put some information in the supplementary. For example, "River, flood wall and discharge data obtained from the Shanghai Municipal Water Authority, to develop the urban inundation model", it is too simple.*
**Response:** Thank you for suggestions. The model utilized in this study integrates rainfall and storm surge boundary conditions, incorporating flood walls along the Huangpu River in Shanghai, as developed and validated by Ke et al. (2021). Digital elevation model data obtained from the Institute of Geographic Sciences and Natural Resources Research, Chinese Academy of Sciences, along with river, flood wall, and discharge data from the Shanghai Municipal Water Authority, were utilized to develop the urban inundation model. The overland flooding model employs a combination of regular rectangular and irregular triangular meshes, with a mesh grid resolution set at 150 m. Storm surge data from the Wusongkou gauge serve as the downstream boundary condition for the overland inundation model. Hourly rainfall hyetograph data sourced from the CMA are inputted into the overland inundation model to ascertain the spatial and temporal distribution of rainfall across Shanghai. For further details regarding the inundation model process for compound flooding, please refer to Ke et al. (2021).

References:
Ke, Q., Yin, J., Bricker, J., Savage, N., Buonomo, E., Ye, Q., Visser, P., Dong, G., Wang, S., Tian, Z. and Sun, L.: An integrated framework of coastal flood modelling under the failures of sea dikes: a case study in Shanghai. Nat. Hazards, 109(1), 671-703, doi:10.1007/s11069-021-04853-z, 2021.

---

## Author Response (AR2)

**Response to Editor's Comments and Suggestions**

*1. Consider adding the reference to the Section number (e.g., "2.1") in the figure where the processes are described.*

**Response:** Thank you for your suggestions. We add the references of different method section numbers in the Figure 1 where the processes described. The updated figure as follows:

[Figure]

Figure 1: Flowchart of this study.

*2. Two points here:- Maybe explain in a sentence why the SFA is needed to begin with. - The way the equation is presented, I would expect A to linearly multiply V0 in time, resulting in a constant shift of the value of interest. But this is not what is visitable in Figure 3. So I assume A is not a linear constant multiplication?*

**Response:** Thank you for your valuable comments. To clarify the Same Frequency Amplification (SFA) method, we added the followings: "The SFA enables the consistent amplification of input variables across different frequency bands, ensuring that the derived signals maintain their original characteristics while enhancing their magnitudes for more accurate modeling and analysis." We agree with your comments and in the revised manuscript, we clarified that $A$ is a constant factor that influences the value of storm surge and rainfall in a linear manner. We included a more detailed explanation in the manuscript as follows: "$A$ represents a constant amplification factor that interacts with $v_0(t)$ in a linear manner. It is the ratio between the potential designed value calculated by the Copula model and the observed values from the 9711 compound flooding events (Figure 3)."

*3. I couldn't find information on the copula model in the manuscript. Which copula is used? How was it set? variables, uncertainty... Some further information is needed.*

**Response:** Thank you for your insightful comment about the Copula model. We built the Copula model via 210 compound flooding events caused by storm surge and rainfall in Shanghai local. In the manuscript, we described how the copula was set up, including the process of selecting the appropriate copula type and estimating its parameters. The revised manuscript is "We employed the Gaussian Copula, which is well-suited (Maximum Likelihood Estimation) for capturing the dependency via 210 compound flooding events caused by storm surge and rainfall in Shanghai area. These design values derived from the probabilistic dependence model presented in Xu et al., 2022. Readers are referred to (Xu et al., 2022) for details on the selection of the severity and magnitude of the event of interest."

References:
Xu, H., Tian, Z., Sun, L., Ye, Q., Ragno, E., Bricker, J., Mao, G., Tan, J., Wang, J., and Ke, Q.: Compound flood impact of water level and rainfall during tropical cyclone periods in a coastal city: the case of Shanghai, *Nat. Hazards Earth Syst. Sci.*, 22(7), 2347-2358, doi:10.5194/nhess-22-2347-2022, 2022.

*4. I couldn't find any reference to figure 8 in the text. Either refer to it or remove the figure.*

**Response:** Thank you for pointing out missing reference for Figure 8. We revised the manuscript to include a reference to Figure 8 at the end of second paragraph in section 5.

*5. To meet with FAIR, and for the benefits of the readers, I would suggest having your copula-SFA model available (with an example) online, via Zenodo or other archive platform. This is not a most but highly suggested.*

**Response:** Thank you for your valuable suggestion regarding the availability of our Copula-SFA model. We provide the model along with example on 4TU (https://data.4tu.nl/datasets/4ff14dc0-a290-4ffd-985f-3d68b9c25644/1). Thank you once again for your constructive suggestion.